# The Huntington’s Disease Gene in an Italian Cohort of Patients with Bipolar Disorder

**DOI:** 10.3390/genes14091681

**Published:** 2023-08-25

**Authors:** Camilla Ferrari, Elena Capacci, Silvia Bagnoli, Assunta Ingannato, Sandro Sorbi, Benedetta Nacmias

**Affiliations:** 1Department of Neuroscience, Psychology, Drug Research and Child Health (NEUROFARBA), University of Florence, 50139 Florence, Italy; elena.capacci@unifi.it (E.C.); silvia.bagnoli@unifi.it (S.B.); assunta.ingannato@unifi.it (A.I.); sandro.sorbi@unifi.it (S.S.); benedetta.nacmias@unifi.it (B.N.); 2IRCCS Fondazione Don Carlo Gnocchi, 50143 Florence, Italy

**Keywords:** Huntington’s disease, CAG, *HTT* gene, intermediate allele, bipolar disorder, psychiatric disorder

## Abstract

Background and objectives: Huntington’s disease (HD) is characterized by motor, cognitive and psychiatric manifestations and caused by an expansion of CAG repeats over 35 triplets on the huntingtin (*HTT*) gene. However, expansions in the range 27–35 repeats (intermediate allele) can be associated with pathological phenotypes. The onset of HD is conventionally defined by the onset of motor symptoms, but psychiatric disturbances can precede the motor phase by up to twenty years. The aims of the present study are to identify HD patients in the pre-motor phase of the disease among patients diagnosed with bipolar disorders and evaluate any differences between bipolar patients carrying the normal *HTT* allele and patients with the expanded *HTT* gene. Methods: We assessed the *HTT* genotype in an Italian cohort of 69 patients who were affected by either type 1 or type 2 bipolar disorder. Results: No patient was found to be a carrier of the pathological *HTT* allele, but 10% of bipolar subjects carried an intermediate allele. Carriers of the intermediate allele were older at the onset of psychiatric symptoms than non-carriers. Conclusion: The pathological *HTT* gene was not associated with bipolar disorder, while we found a higher frequency of the intermediate allele among the bipolar population with respect to healthy controls. The identification of this subset of bipolar subjects has implications for the clinical management of patients and their family members and promotes further investigation into possible pathological mechanisms common to both HD and bipolar disorder.

## 1. Introduction

Expansion of CAG triplets in the huntingtin gene (*HTT*) on chromosome 4 is responsible for Huntington’s disease (HD). The presence of 40 or more CAG triplets is invariably associated with manifest disease, while expansions in the range 36–39 repeats are considered to confer a reduced disease penetrance [1,2]. HD is characterized by motor, cognitive and psychiatric disorders [1]. Conventionally, HD onset is defined as the age of the onset of motor signs; however, psychiatric disturbances, especially mood disturbances, can often occur up to twenty years before the manifest motor phase [3,4]. Observational studies report psychiatric symptoms to be the first manifestation of HD in about 20% of cases [5].

HD is transmitted in an autosomal dominant manner, but new mutations can be generated in the offspring via the elongation of unstable alleles falling in the range 27–35 repeats, which are defined as intermediate alleles (IA) [6]. The percentage of IA has been variably estimated to be between 0.45 and 8.7% [7,8] in the healthy population. The reported frequency of IA in European-based cohorts is about 6% [7]. Although the *HTT* gene with less than 36 CAG repeats is considered to be normal, a growing number of studies have reported that subjects carrying IA are at higher risk of behavioral problems and can even develop the clinical phenotype of HD [9,10,11,12,13]. Bipolar disorder is a multifactorial disease with a high heritability rate, which is estimated at 60–90% in twin studies. However, the genetic architecture of the disorder is still far from being elucidated [14,15]. Hypomanic and manic episodes have been reported in up to 10% of HD patients [16,17], and the pathogenesis of HD and bipolar disorder is connected to the same brain structures [18,19,20,21,22,23]. This finding could suggest the presence of common mechanisms underlying the development of HD and bipolar disease. Recent studies have demonstrated that the length of CAG repeats can influence the risk of the development of depressive symptoms [24], and major depression could be a pre-motor phenotype of HD [25]. Subjects with expanded *HTT* alleles and pre-motor psychiatric symptoms could be misdiagnosed as having primary psychiatric disease, while their early identification would imply different clinical management, such as specific clinical follow-up, tailored treatment [26] and the genetic counseling of family members [6,11,13]. In the present study, we analyzed the *HTT* gene in a psychiatric cohort of patients affected by bipolar disorder in order to identify pre-motor HD patients and evaluate whether the length of CAG repeats may influence or contribute to the development of bipolar disorder.

## 2. Materials and Methods

### 2.1. Population

The blood samples of 69 unrelated patients affected by bipolar disorder type I and type II were genotyped to determine the presence of the *HTT* gene. The diagnosis of bipolar disorder was made according to the DSM IV criteria [27]. 

Demographic–clinical data included the following information: age at the time of withdrawal, age at disease onset, gender, first symptom at onset, educational level and family history of psychiatric disorders. The genetic results were compared to a cohort of 104 healthy controls that we recently described, i.e., 45 males (43.3%) and 59 females (56.7%), with the mean age being 64.4 (±8.5) years old [28].

### 2.2. Genetic Analysis

Genomic DNA was isolated from peripheral blood samples using a QIAamp DNA Blood Mini QIAcube kit (cat. No 51126 QIAGEN) and quality-checked via a QIAxpert spectrophotometer. DNA samples were stored at +4° until use. CAG repeat expansion of the *HTT* gene was investigated via a polymerase chain reaction (PCR) amplification assay, using the primers 5′-[6-FAM] GACCCTGGAAAAGCTGATGA-3′ and 5′-GGCTGAGGAAGCTGAGGAG-3′. The forward primer was modified using the fluorescent dye 6-carboxyfluorescein (6-FAM) [29].

The PCR was performed using 10 ng of genomic DNA, a 1xTaq MegamùMix (Microzone with Buffer master mix) and 1.5 µL of each primer and water up to a final volume of 15 µL. The PCR was performed via 35 cycles of 30 s denaturation at 94 °C, 30 s of annealing at 60 °C, 30 s of elongation at 72 °C and 2 min of elongation at 74 °C. Every PCR included a negative control without genomic DNA and two positive control samples with predetermined 19/20 and 23/27 *HTT* CAG repeats. The PCR products were run via a capillary electrophoresis using the SeqStudio Genetic Analyzer (ThermoFisher, Monza, Italy) and analyzed using the GeneMapper software (v.4.0, Applied Biosystems, Waltham, MA, USA). A set of known length *HTT* CAG alleles was used as the size standard. A CAG trinucleotide expansion under 27 repetitions was considered to be normal allele, IAs with repeats were in the range 27–35 and pathologic alleles had expansion sizes >35 repeats.

### 2.3. Statistical Analysis

The demographic–clinical features of bipolar patients were described via the mean and standard deviation in case of continuous variables and via percentages in the case of categorical variables. Due to the identification of IA, we performed a comparison between bipolar patients carrying IA (IA carriers) and bipolar patients carrying the normal alleles (non-IA carriers). The comparisons between the demographic–clinical features were performed through non-parametric analysis: Fisher’s exact test was used for categorical variables and Mann–Whitney test was used for continuous variables. The study population was then compared to the healthy control group in terms of the *HTT* CAG distribution and IA frequency. The Mann–Whitney test and Fisher’s exact test were used if appropriate. Statistical analyses were performed using IBM SPSS Statistics 20.0 (IBM Corp., New York, NY, USA).

## 3. Results

There were 69 bipolar patients with a mean age at disease onset of 34 years old and a tendency to be female (60.9%) (Table 1). The onset of bipolar disorder occurred in 49.2% of cases with depressive symptoms, in 15.9% of cases with maniacal symptoms and in the remaining cases with a mixture of both symptoms. More than 60% of patients had type 1 bipolar disorder. A family history of psychiatric disorders was present in 43.5% of cases. The mean number of CAG repeats in the *HTT* gene was 17.7 for the shorter allele and 21.03 for the longer allele (Table 1). No patient was found to be a carrier of a pathological allele. We identified seven bipolar patients carrying IA (IA carriers). The mean length of IA was 28.6 (ranging 27–33). The comparison between bipolar patients IA carriers and non-IA carriers is reported in Table 2. IA carriers were older at disease onset than non-IA carriers (Table 2). The frequency of IA was statistically significant higher in bipolar patients, i.e., 10.14% (7 out of 69), than in the healthy control population, i.e., 5.8% (6 out of 104) (*p* < 0.001), while the range of normal CAG repeats was 13–26 in both groups. The distribution of the *HTT* CAG repeat length among the two cohorts is shown in Figure 1. The distribution of the normal-range alleles differs in the two cohorts, as healthy subjects have higher numbers of triplets in both alleles (Figure 1).

## 4. Discussion

The study of the *HTT* gene in a cohort of 69 patients with bipolar disorder did not identify any subjects in a pre-motor stage of HD; in fact, no patient was a carrier of the pathological allele. The IA was detected in 10.14% of cases. IA carriers were older than non-IA carriers at the disease’s onset. Although psychiatric symptoms may often precede the motor phase of HD, only three previous studies evaluated the frequency of the pathological expansion in the *HTT* gene in psychiatric cohorts [24,25,30]: two studies in patients affected by major depression [24,25] and one study in a Brazilian cohort affected by bipolar disorder [30]. Major depression was frequent in the pre-motor phase of HD [25], while Ramos et al. [30], in line with our results, did not identify any association between the pathological *HTT* allele and bipolar disorder. The frequency of IA carriers in our psychiatric cohort was almost the double that detected in the local reference population. Data from the literature have already shown a higher risk of behavioral disturbances in subjects carrying IA [10], such as apathy, obsessive disorder, anxiety, depression and suicidal ideation. However, the mechanism through which the IA of *HTT* gene is associated with psychiatric manifestations remains unclear [10]: one hypothesis states that the association occurs through mechanisms unrelated to those leading to HD pathology, such as interaction with other genes [10]. In that case, the difference in the genetic structure between ethnic groups could be explained based on the lack of association between IA and bipolar disorder in the Brazilian population [30,31]. The analysis of the distribution of the *HTT* normal alleles revealed a statistically significant difference between bipolar patients and controls, with shorter normal alleles among patients. These data support the hypothesis of a non-linear association between the CAG repeat length and the risk of bipolar disorder, which means that both the normal alleles in the shorter range and the IA could increase the risk of developing bipolar disorder. This non-linear association between the *HTT* CAG length and the risk of developing major depression was previously described [24], as it was between the *HTT* size and intelligence in the general population [32]. Therefore, it seems that the increasing size of the non-pathological *HTT* gene confers an advantage up to the values that determine pathology. In fact, the increasing number of *HTT* triplets is part of the evolutionary process [33,34,35], and the variation in the number of repeats across the normal range influences the brain’s structure in healthy subjects [36]: longer non-pathological alleles have been found to be associated with an increasing volume of basal ganglia [36]. Studies of HD patients [37,38,39] demonstrated that the increasing sizes of normal alleles could be protective by mitigating the effects of the pathological allele and delaying disease onset. Interestingly, in the present cohort, we found that bipolar patients carrying IA were older at disease onset than non-IA carriers. Basal ganglia are also involved in the pathophysiology of bipolar disorder [20,40,41]; thus, IA, via its effect on basal ganglia [36], could have a protective role and delay the onset of bipolar symptoms. 

Due to the multiplicity of functions of the *HTT* gene, the neuroprotective effect of IA could also depend on other mechanisms: the regulation of genes transcription, the modulation of synaptic transmission and the anti-apoptotic and anti-oxidant functions [23,34].

The number of *HTT* triplets may also modulate the individuals’ predisposition to the effects of other genetic and environmental factors. 

Another hypothesis could be that when the development of bipolar symptoms is driven by the IA of *HTT*, the age at onset is later than when it is caused by other genetic factors.

The main limitations of the present study are the small sample sizes used, the mono-centric setting and the lack of information about the genetic status of the patients’ family members. Thus, the generalizability of the results is questionable, and the interpretation of the findings, while intriguing, can only be speculative. The suggestions that the IA of *HTT* plays a role in the development of bipolar disorder or is associated with a peculiar new non-motor phenotype in the spectrum of HD disorder are both fascinating hypotheses that need to be clarified through a larger study of bipolar patients and their relatives. Regardless, beyond the interpretation of the role played by the *HTT* gene in the development of bipolar symptoms, the identification of IA carriers among this Italian cohort of bipolar patients has important implications for both patients and their families: patients can be at risk of developing a complete HD phenotype [11,13], and family members should be made aware of their increased risk of developing behavioral disturbances [10] or inheriting a pathological expanded *HTT* gene [6]. 

## 5. Conclusions

To the best of our knowledge, there is only one other study that evaluated the *HTT* genotype among bipolar patients [26], and similar to our study, bipolar disorder was not associated with the pathological *HTT* allele. In our cohort, IA was associated with bipolar disorder, albeit with a delayed age at onset. The findings of the present brief report are intended to be proof of concept that justify further investigation into the role played by the IA of the *HTT* gene in the onset of bipolar disorder.

## Figures and Tables

**Figure 1 genes-14-01681-f001:**
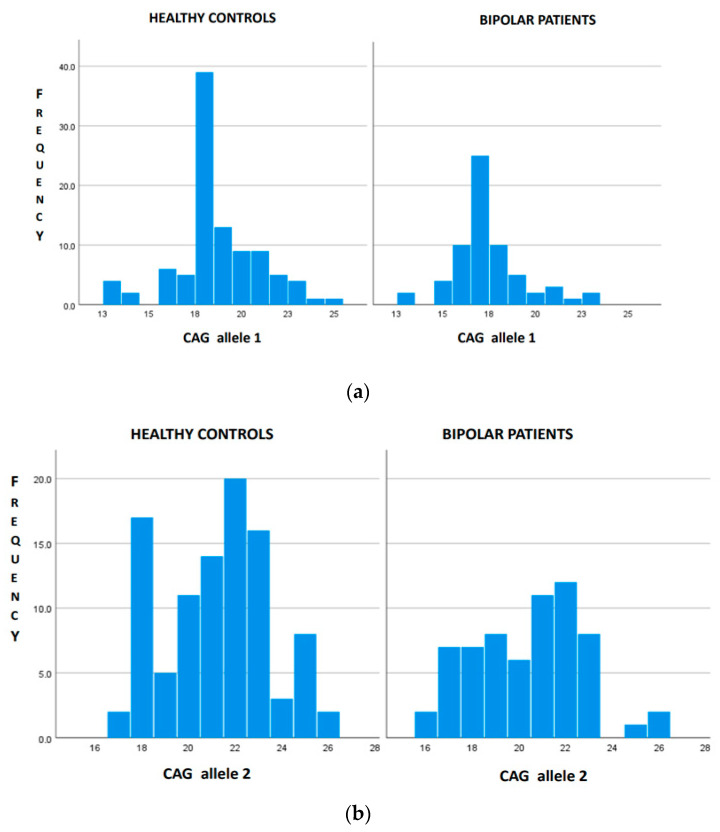
Distribution of *HTT* gene length in bipolar patients (n = 62) and healthy controls (n = 98) with the exclusion of IA carriers. (**a**) The distribution of CAG in allele 1 differs between healthy controls and bipolar patients (*p* < 0.001). (**b**) The distribution of CAG in allele 2 differs between healthy controls and bipolar patients (*p* = 0.028).

**Table 1 genes-14-01681-t001:** Clinical–demographic and genetic features of the studied patients.

	Patients (n = 69)
Age (SD)	53.91 (10.2)
Age at disease onset (SD)	34.81 (13.4)
Sex	
Male (%)	27 (39.1)
Female (%)	42 (60.9)
Education in years (SD)	10.25 (3.1)
Symptoms at onset	
Maniacal (%)	11 (15.9)
Depression (%)	34 (49.2)
Mix (%)	24 (34.7)
Type 1 bipolar disorder (%)	44 (63.7)
Family history of psychiatric disorders (%)	30 (43.5)
*HTT* gene *	
- Allele CAG 1 (SD)	17.7 (2.1)
- Allele CAG 2 (SD)	21.03 (3.3)
- -------------------------------	-
- Pathological allele *	0
- Intermediate allele (IA) (%)	7 (10.2)

* Pathological allele > 35 CAG repeats; intermediate allele 27–35 CAG repeats; normal allele < 27 CAG repeats; allele CAG 1 = shorter allele; allele CAG 2 = longer allele.

**Table 2 genes-14-01681-t002:** Characteristics of bipolar patients by the genetic status of IA carriers.

	No * IA Carriers (n = 62)	* IA Carriers (n = 7)	*p*
Age at disease onset (SD)	33.7 (13.5)	43.3 (9.9)	0.048
Sex			
Male (%)	23 (37.1%)	4 (57.3%)	0.38
Female (%)	39 (62.9%)	3 (42.8)	
Symptoms at onset			
Maniacal (%)	9 (14.5)	2 (28.5)	0.89
Depression (%)	30 (48.4)	4 (57.1)	
Mix (%)	23 (37.1)	1 (14.2)	
Family history of psychiatric disorder (%)	39 (45.1)	2 (28.5)	0.09
CAG 1 (SD)	17.37 (1.8)	20.6 (2.3)	0.09
CAG 2 (SD)	20.2 (2.08)	28.6 (1.4)	<0.001

* IA = intermediate allele.

## Data Availability

The data of this study are available from the corresponding author, C.F., upon reasonable request.

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
