# Peer review of "The Huntington’s Disease Gene in an Italian Cohort of Patients with Bipolar Disorder"

_genes, 2023, doi:10.3390/genes14091681_

Round 1

Reviewer 1 Report

Overall, the organisation and content of the manuscript are commendable. It follows a logical structure with clear headings for each section Nonetheless, there are a few minor suggestions worth considering for further improvement or clarification.

Abstract Section

The authors should consider adding a brief sentence explaining the potential implications of the higher frequency of intermediate alleles among bipolar patients and how this might contribute to the understanding of the relationship between HD and bipolar disorder.

Introduction Section

The authors could briefly mention why identifying pre-motor HD patients among those with bipolar disorder is important. Is it to provide early intervention or to understand potential shared mechanisms between the two disorders? Also, they could highlight any gaps in the current knowledge that their study aims to address.

Materials and Methods Section

The authors should acknowledge any limitations of their study that may have influenced the results or the interpretation of the findings. This helps provide a balanced perspective and suggests areas for improvement in future studies.

Results Section is well-organised and provides a clear overview of the key findings.

Discussion Section

The authors could explore potential applications of their findings in a clinical setting: How might these findings influence clinical practice?

How might the identification of IA-carriers among bipolar patients influence clinical management or patient counselling?

Could the identification of IA-carriers among bipolar patients have implications for early diagnosis, risk assessment, or personalised treatment strategies?

Author Response

Overall, the organisation and content of the manuscript are commendable. It follows a logical structure with clear headings for each section.

We thank the Reviewer for the general positive comment

Nonetheless, there are a few minor suggestions worth considering for further improvement or clarification.

-Abstract Section

The authors should consider adding a brief sentence explaining the potential implications of the higher frequency of intermediate alleles among bipolar patients and how this might contribute to the understanding of the relationship between HD and bipolar disorder.

As suggested by the reviewer we have added the following sentence to the Abstract: “The identification of this subset of bipolar subjects has implications for the clinical management of patients and their family members and promotes further investigation into possible pathological mechanisms in common between HD and bipolar disorder.”

-Introduction Section

The authors could briefly mention why identifying pre-motor HD patients among those with bipolar disorder is important. Is it to provide early intervention or to understand potential shared mechanisms between the two disorders? Also, they could highlight any gaps in the current knowledge that their study aims to address.

According to the reviewer’ suggestion we have discussed the importance of identifying subject in pre-motor HD phase The identification of this subset of patient is important for the early detection of HD cases, that imply different management of the patients and their family (Introduction Lines 55-58). Moreover the analysis of HTT in bipolar subjects could highlight the presence of genetic connection as the basis of shared mechanisms between HD and bipolar disease. We have provide references supporting the presence of common pathogenetic mechanism in HD and bipolar disease (references 18-23;  Introduction Lines 50-52 )

-Materials and Methods Section

The authors should acknowledge any limitations of their study that may have influenced the results or the interpretation of the findings. This helps provide a balanced perspective and suggests areas for improvement in future studies.

We agree with the reviewer, a paragraph addressing the limitations of the study has been added in the Discussion  (Lines 185-191)

-Results Section is well-organised and provides a clear overview of the key findings.

 We thank the reviewer for the positive feed-back

-Discussion Section

The authors could explore potential applications of their findings in a clinical setting: How might these findings influence clinical practice? How might the identification of IA-carriers among bipolar patients influence clinical management or patient counselling? Could the identification of IA-carriers among bipolar patients have implications for early diagnosis, risk assessment, or personalised treatment strategies?

We agree with the reviewer, the present findings can have immediate implications in clinical practice. Bipolar IA-carriers could represent a new phenotype associated with HTT gene or could be subjects in the pre-motor phase of HD thus they require a close and specific clincal follow-up, a tailored tretament and their family members deserve genetic counseling in relation to their risk of having inherited a pathological HTT allele. We have added these points in the introduction (Lines 55-58) and in the Discussion (Lines 188-196). 

Reviewer 2 Report

The research is fairly sound and has no major shortcomings save the fact that the authors speculate a lot although only 7 patients (out of 69) were IA carriers. The number of participants (69) is barely enough for genetic analysis and I would be very cautious to extrapolate findings from just 7 patients onto the whole BP disorder population. Therefore, the authors must clearly address this issue in the limitations paragraph. Having said that, this may serve as a brief report but nothing else as this topic deserves proper investigation with a sufficient number of participants.

Author Response

The research is fairly sound and has no major shortcomings save the fact that the authors speculate a lot although only 7 patients (out of 69) were IA carriers. The number of participants (69) is barely enough for genetic analysis and I would be very cautious to extrapolate findings from just 7 patients onto the whole BP disorder population. Therefore, the authors must clearly address this issue in the limitations paragraph.

We thank the reviewer for the interest in our manuscript.                                                                                           Following the reviewer suggestion we have added a section on the limitations of the study (Lines 185-191) addressing that the small sample size, the mono-centric setting and the lack of genetic information on the genetic status of patients’ family member do not allow us to confirm our hypotheses which remain speculative.

Having said that, this may serve as a brief report but nothing else as this topic deserves proper investigation with a sufficient number of participants.

We specified in the Conclusion that this is a brief report which sets the stage for future studies (Lines 201-203)

Reviewer 3 Report

The authors presented a nice and well-written original manuscript entitled "The Huntington's disease gene in an Italian cohort of patients with bipolar disorder". Although not representing the first populational screening study of pathological trinucleotide repeat expansion in a specific neurological scenario, this original study provided interesting aspects looking for patients which could be in the premotor stage of Huntington's disease - which were absent in the sample; and looking for individuals at the intermediate allele interval and describing clinical aspects related to these individuals. Despite not expanding genetic evaluation for other genes previously related to the bipolar disease presentation in patients with negative screening in the population, the manuscript brings interesting contributions to the current literature. Some points could be evaluated by the authors at this point: 

1. The name of human genes in the text should be presented in italics (e.g. HTT). 

2. More than 40% of cases in the studied sample had a positive family history for psychiatric diseases. Have the authors observed if these cases were associated with the intermediate alleles? This is an interesting aspect because it could represent a possible expansion of the pre-expansion (full-expansion) phenotypes or even represent a context of a new phenotype related to the gene (a non-motor and purely psychiatric presentation). 

3. What do authors consider that could modulate the later age at onset of symptoms of patients with IA-carriers? Could intermediate expansion represent a possible neuroprotective factor in this context? Is the basal ganglia pathophysiological involvement possibly related or are there other brain regions (or even other genetic mechanisms) which could have contributed? 

Author Response

The authors presented a nice and well-written original manuscript entitled "The Huntington's disease gene in an Italian cohort of patients with bipolar disorder". Although not representing the first populational screening study of pathological trinucleotide repeat expansion in a specific neurological scenario, this original study provided interesting aspects looking for patients which could be in the premotor stage of Huntington's disease - which were absent in the sample; and looking for individuals at the intermediate allele interval and describing clinical aspects related to these individuals. Despite not expanding genetic evaluation for other genes previously related to the bipolar disease presentation in patients with negative screening in the population, the manuscript brings interesting contributions to the current literature.

We thank the reviewer for the general positive outcome and the interest in our manuscript

Some points could be evaluated by the authors at this point: 

  1. The name of human genes in the text should be presented in italics (e.g. HTT). 

HTT has been changed through the manuscript

  1. More than 40% of cases in the studied sample had a positive family history for psychiatric diseases. Have the authors observed if these cases were associated with the intermediate alleles? This is an interesting aspect because it could represent a possible expansion of the pre-expansion (full-expansion) phenotypes or even represent a context of a new phenotype related to the gene (a non-motor and purely psychiatric presentation). 

We agree with the reviewer on the importance of this information. Unfortunatly genetic data on family members were not available. We have addressed this point as one of the limitations of the study in the Discussion  (Lines 185-188).

  1. What do authors consider that could modulate the later age at onset of symptoms of patients with IA-carriers? Could intermediate expansion represent a possible neuroprotective factor in this context? Is the basal ganglia pathophysiological involvement possibly related or are there other brain regions (or even other genetic mechanisms) which could have contributed?

We thank the reviewer for suggesting to we broaden the discussion on the role of IA in the development of bipolar symptoms. We believe that IA could exert a neuroprotective effect, delaying bipolar disease onset, via different mechanisms, the increased volume of basal ganglia, the interaction with genes, the regulation of synaptic trasmission, the anti-apoptotic and anti-oxidant functions.  We have added these points in the Discussion (Lines 174-181).